# Numerical Analysis of Transient Pressure Damping in Viscoelastic Pipes at Different Water Temperatures

**DOI:** 10.3390/ma15144904

**Published:** 2022-07-14

**Authors:** Qiang Sun, Zhilin Zhang, Yuebin Wu, Ying Xu, Huan Liang

**Affiliations:** 1School of Civil Engineering, Institute of Artificial Environment Control and Energy Application, Northeast Forestry University, Harbin 150040, China; sunqiang@nefu.edu.cn (Q.S.); zhangzhilin@nefu.edu.cn (Z.Z.); 2School of Architecture, Harbin Institute of Technology, Harbin 150090, China; 3Laboratory of Cold Region Urban and Rural Human Settlement Environment Science and Technology, Harbin Institute of Technology, Harbin 150090, China; 4School of Energy and Architecture Engineering, Harbin University of Commerce, Harbin 150028, China; joexying@126.com; 5Industrial Control Energy Saving Business Division, Beijing Huada Zhibao Electronic System Co., Ltd., Beijing 100020, China; lianghuan0720@163.com

**Keywords:** energy analysis, one-dimensional friction model, peak pressure damping, transient flow, viscoelastic pipe

## Abstract

Water temperature affects the peak pressure damping of transient flows in viscoelastic pipes. Owing to the viscoelastic properties of pipes, the accuracy of peak pressure damping simulations hinges on both viscoelastic and frictional factors. In simulations, the influence of both factors on peak pressure damping at different water temperatures is unclear. In this study, the Kelvin–Voigt model with both a quasi-steady friction model and modified Brunone model was employed. Based on experimental data, the accuracy of simulated peak pressure damping was verified at four different water temperatures (13.8, 25, 31, and 38.5 °C). From the perspective of energy transfer and dissipation, the influence of viscoelastic and frictional factors on peak pressure damping were clarified, and the applicability of different friction models was determined based on the contributions of viscoelastic and frictional factors to peak pressure damping. The numerical results indicate that the viscoelastic properties of pipes have a greater impact on peak pressure damping than their frictional properties at 25, 31, and 38.5 °C. Higher temperatures result in a delay in the rate of work and a decrease in the frequency of work performed by viscoelastic pipes. Viscoelastic properties play a more important role than frictional ones in calculating peak pressure damping as the water temperature increases. In addition, the one-dimensional quasi-steady friction model can accurately simulate peak pressure damping within a specified water temperature range.

## 1. Introduction

Viscoelastic pipes such as those made of polyvinyl chloride (PVC), low-density polyethylene (LDPE), and high-density polyethylene (HDPE) are widely used in urban supply systems. When valves suddenly close or pumps stop abruptly, the resultant water hammer phenomenon can threaten the safe operation and maintenance of a hydraulic system. Accurately simulating this phenomenon in viscoelastic pipes is thus very important [1]. Both friction and viscoelasticity have significant impacts on the calculation of the peak pressure damping of transient flows in viscoelastic pipes, especially at different water temperatures [2]. This is because the water temperature affects not only the fluid characteristics [3], such as the density and viscosity of the water, but also the range of linear viscoelasticity with respect to the viscoelastic pipes [4,5,6]. Therefore, it is necessary to investigate the influence of viscoelastic and frictional factors in the numerical simulation of the peak pressure damping of transient flows in viscoelastic pipes at different water temperatures.

Several numerical models have been used to investigate friction and viscoelasticity. With respect to viscoelasticity, the Kelvin–Voigt (K–V) model has been used to describe the viscoelastic behavior of the pipe wall [7]. As for friction, one-dimensional (1D) quasi-steady and unsteady friction models [8,9,10,11] and two-dimensional friction models [12,13,14,15] have been used to describe the wall shear stress in transient flows. Generally, the classic transient flow model with a quasi-steady friction model is capable of accurately simulating the maximum value in terms of the pressure fluctuation in elastic pipes; however, it cannot accurately describe the peak pressure damping in most instances [1]. Cao et al. [8] modified the 1D instantaneous acceleration-based (IAB) model to better predict the wave peak and valley. Urbanowicz [9] improved the 1D model with a weighting function to calculate the unsteady-state component. Pezzinga et al. [10] compared the numerical results of 1D elastic and viscoelastic models used to simulate transient flows in HDPE pipe systems. The results showed that the viscoelastic model was more accurate when simulating the peak pressure damping than the elastic model. Abdeldayem et al. [11] compared the accuracy of different unsteady fiction models in engineering practice. They concluded that the modified IAB model is relatively suitable for simulating pressure fluctuation in transient flows. Covas et al. [16] investigated the effects of the viscoelastic parameters of the K–V model and verified the accuracy of the transient flow. Duan et al. [17,18] discussed the characteristics of viscoelastic pipes under transient flow with respect to energy transfer and dissipation based on energy analysis.

Another important factor that affects the peak pressure damping of transient flows is friction. Duan et al. [19] investigated the influence of the scale of the pipe system, particularly the pipe length and diameter, on the unsteady friction of transient flows in elastic pipes. The numerical results showed that a higher scale ratio between the pipe length and pipe diameter resulted in an increase in the role of unsteady friction on the peak pressure damping of transient flows in elastic pipes. Based on the experimental data reported by Covas et al. [14], Seck et al. [20] modified the unsteady friction correlation coefficient to improve the accuracy of the unsteady friction model. The results showed that the inclusion of viscoelasticity and unsteady friction generated dramatic peak pressure damping.

The water temperature affects not only the viscoelasticity of the pipe but also the speed and attenuation of the pressure wave of the transient flow in viscoelastic pipes. Neuhaus et al. [21] conducted transient flow experiments at different temperatures and compared the accuracy of simulation results under different tests. Saidani et al. [22] investigated the effect of temperature on a transient cavitation flow. Mousavifard [23] studied the influence of temperature on turbulence parameters during transient cavitation flows in viscoelastic pipes. The results showed that at higher temperatures, the flexibility of the pipe and the delay of the velocity distribution increased, whereas the velocity gradient near the wall decreased. 

As mentioned above, some researchers have explored the effects of viscoelasticity and friction on the peak pressure damping of transient flows in viscoelastic pipes; however, the influence of viscous-elasticity and friction on peak pressure damping at different water temperatures has not yet been thoroughly studied. Therefore, this study analyzed simulation results for the peak pressure damping of transient flows in viscoelastic pipes at different water temperatures; the simulations employed the modified K–V model with the quasi-steady friction model and modified Brunone model. The mean absolute percentage error (MAPE) and difference evaluation methods were used to estimate the accuracy of the numerical simulations of the peak pressure damping. Then, from the perspective of energy transfer and dissipation, the influence of friction and viscoelasticity on peak pressure damping was investigated. The contributions of thviscoelastic, quasi-steady friction, and unsteady friction factors to the peak pressure damping in viscoelastic pipes are discussed separately.

## 2. Establishment of Mathematical Models

### 2.1. Governing Equations

The 1D transient flow equation consists of continuity and momentum equations, which can be expressed in the following form [14]:(1)$$\frac{\partial H}{\partial t}+\frac{a^{2}}{gA}\frac{\partial Q}{\partial x}+\frac{2a^{2}}{g}\frac{\partial \varepsilon_{r}}{\partial t}=0$$
(2)$$\frac{\partial Q}{\partial t}+gA\frac{\partial H}{\partial x}+\frac{\pi D\tau_{w}}{\rho }=0$$
where *H* is the piezometric head, *Q* is the flow rate, *a* is the pressure wave speed, *A* is the pipe cross-sectional area, *x* is the coordinate along the pipeline axis, *t* is the time, *τ_w_* is the wall shear stress, and *ε_r_* is the retarded strain. The dependent variables are *H*, *Q*, *τ_w_*, and *ε_r_*, which are functions of *x* and *t*, respectively. 

In Equations (1) and (2), the time derivative of the retarded strain, $\frac{\partial \varepsilon_{r}(t)}{\partial t}$, and the wall shear stress, *τ_w_*, cannot be calculated directly; thus, further numerical discretization is required.

#### 2.1.1. Numerical Discretization of the Wall Shear Stress

The wall shear stress, *τ_w_*, is defined as having two parts in transient flows: *τ_q_* is the steady-state component, and *τ_u_* is the unsteady-state component [1].
(3)$$\tau_{w}=\tau_{q}+\tau_{u}$$

For the quasi-steady friction (QF) model, the steady-state component, *τ_q_*, is calculated as follows [4]:(4)$$\tau_{q}(t)=\frac{f}{8}\rho V^{2}$$
where *V* is the average velocity in the section, *f* is the Darcy–Weisbach friction factor, and *ρ* is the density of the liquid.

The unsteady-state component, *τ_u_*, is typically neglected in the QF model. However, rapid transient events and high pulsating frequencies require the accurate representation of unsteady friction [24,25,26,27]. 

For the unsteady friction (UF) model, the modified Brunone model was employed in this study. It is assumed that instantaneous local acceleration and convective acceleration affect unsteady friction [28,29,30]. Hence, the unsteady-state component is calculated as follows:(5)$$\tau_{u}(t)=\frac{k^{*}\rho D}{4}\left(\frac{\partial V}{\partial t}+sign(V)a\left|\frac{\partial V}{\partial x}\right|\right)$$
where ∂V/∂x is calculated as follows:(6)$$\frac{\partial V}{\partial x}=\frac{\Delta V}{\Delta x}$$

Through theoretical analysis of simulation results and experimental data, Vardy and Brown [27] provided a method to calculate the value of the empirical coefficient of the Reynolds number as follows:(7)$$k^{*}=\sqrt{C^{*}}/2$$
(8)C*=0.00476Re≤23007.41/Relg(14.3/Re0.05)Re≥2300

#### 2.1.2. Numerical Discretization of the Retarded Strain

The Kelvin–Voigt (K–V) model (Figure 1) is the most commonly used model to describe the creep compliance and relaxation modulus of materials. The creep function can be expressed as follows [31]:(9)$$J(t)=J_{0}+\sum_{k=1}^{N}J_{k}(1-e^{-t/\tau_{k}})$$
where *J*_0_ = 1/*E*_0_ is the instantaneous elastic creep, *J_k_* = 1/*E_k_* is the creep compliance of the spring of the *k*th element, *τ_k_* = *F_k_*/*E_k_* is the retarded component of the *k*th element, *F_k_* is the viscosity of the *k*th element, *E_k_* is the modulus of elasticity of the spring of the *k*th element, and *N* is the number of K–V elements.

The total strain, *ε*, is given by the sum of the instantaneous strain, *ε_e_*, and the retarded component, *ε_r_* [31].
(10)$$\varepsilon =\varepsilon_{e}+\varepsilon_{r}$$

Using the K–V model with *N* elements, the retarded strain is the sum of the single-element deformation [16].
(11)$$\partial \varepsilon_{r}(t)=\sum_{k=1\cdots N}\partial \varepsilon_{rk}(t)$$

According to the Boltzmann superposition principle [32], the time derivative of the retarded strain in Equation (1) is expressed as follows:(12)$$\frac{\partial \varepsilon_{r}(t)}{\partial t}=\sum_{k=1\cdots N}\frac{\partial \varepsilon_{rk}(t)}{\partial t}$$

Considering the creep function (Equation (9)), the retarded strain can be written as follows [31]:(13)$$\varepsilon_{r}(x,t)=\sum_{k=1\cdots N}\varepsilon_{rk}(x,t)=\sum_{k=1\cdots N}\frac{\gamma \alpha D}{2e}\int_{0}^{t}\left[H(x,t-t^{\prime })-H_{0}(x)\right]\frac{J_{k}}{\tau_{k}}e^{\frac{-t^{\prime }}{\tau_{k}}}dt^{\prime }$$
where γ is the bulk weight.

### 2.2. Numerical Scheme

The governing equations are solved using the method of characteristics (MOC). This method transforms the partial differential equations into ordinary differential equations by calculating along the characteristic lines *dx*/*dt* = ±*a*, as shown in Figure 2. The ordinary differential equations are given as follows:(14)$$\frac{dH}{dt}\pm \frac{a}{gA}\frac{dQ}{dt}\pm \frac{a}{g}\frac{4\tau_{w}}{\rho D}+\frac{2a^{2}}{g}\frac{\partial \varepsilon_{r}}{\partial t}=0$$

By integrating Equation (14) on the characteristic lines between times *n*Δ*t* and (*n* + 1)Δ*t*, the discretized forms can be obtained as follows:(15)$$H(x,t)-H(x\pm \Delta x,t-\Delta t)\mp \frac{a}{gA}[Q(x,t)-Q(x\pm \Delta x,t-\Delta t)]\mp \frac{a}{g}\frac{4\tau_{w}(x,t)}{\rho D}+\frac{2a^{2}}{g}\frac{\partial \varepsilon_{r}(x,t)}{\partial t}=0$$

By substituting Equation (13) into Equation (15), the characteristic equations can be obtained as follows:(16)$$(1+\frac{2a^{2}\Delta t}{g}\sum_{k=1}^{N}\frac{\alpha D}{2e}\gamma \frac{J_{k}}{\Delta t}e^{-\Delta t/\tau_{k}})H_{i}^{n+1}+\frac{a}{gA}Q_{i}^{n+1}+\frac{4a\Delta t}{\rho gD}\tau_{w,i-1}^{n}=C_{p}$$
(17)$$(1+\frac{2a^{2}\Delta t}{g}\sum_{k=1}^{N}\frac{\alpha D}{2e}\gamma \frac{J_{k}}{\Delta t}e^{-\Delta t/\tau_{k}})H_{i}^{n+1}-\frac{a}{gA}Q_{i}^{n+1}-\frac{4a\Delta t}{\rho gD}\tau_{w,i-1}^{n}=C_{m}$$
where the coefficients *C_p_* and *C_m_* are written as
(18)$$C_{p}=H_{i-1}^{n}+\frac{a}{gA}Q_{i-1}^{n}-\frac{4a\Delta t}{\rho gD}\tau_{w,i-1}^{n}+\frac{2a^{2}\Delta t}{g}\sum_{k=1}^{N}V_{E}$$
(19)$$C_{m}=H_{i+1}^{n}-\frac{a}{gA}Q_{i+1}^{n}+\frac{4a\Delta t}{\rho gD}\tau_{w,i+1}^{n}+\frac{2a^{2}\Delta t}{g}\sum_{k=1}^{N}V_{E}$$
(20)$$V_{E}=\frac{\alpha D}{2e}\gamma \frac{J_{k}}{\Delta t}e^{-\Delta t/\tau_{k}}H_{0}+\frac{\alpha D}{2e}\gamma \frac{J_{k}}{\Delta t}(1-2e^{-\Delta t/\tau_{k}})H(x,t-\Delta t)+\frac{e^{-\Delta t/\tau_{k}}}{\tau_{k}}\varepsilon_{rk}(x,t-\Delta t)$$

By combining Equations (16) and (17), the pressure head, discharge, and retarded strain at (*n* + 1)△*t* can be obtained as follows:(21)$$H_{i}^{n+1}=\frac{C_{p}+C_{m}}{2(1+\frac{2a^{2}\Delta t}{g}\sum_{k=1}^{N}\frac{\alpha D}{2e}\gamma \frac{J_{k}}{\Delta t}e^{-\Delta t/\tau_{k}})}$$
(22)$$Q_{i}^{n+1}=\frac{gA}{2a}(C_{p}-C_{m})$$
(23)$$\varepsilon_{rki}^{n+1}=\gamma \tau_{k}\{H_{i}{}^{n+1}[\frac{\alpha D}{2e}\frac{J_{k}}{\tau_{k}}e^{-\frac{\Delta t}{\tau_{k}}}-\frac{\alpha D}{2e}\frac{J_{k}}{\Delta t}(1-e^{\frac{\Delta t}{\tau_{k}}})]+H_{i}{}^{n}[\frac{\alpha D}{2e}\frac{J_{k}}{\Delta t}(1-e^{\frac{\Delta t}{\tau_{k}}})-\frac{\alpha D}{2e}\frac{J_{k}}{\tau_{k}}e^{-\frac{\Delta t}{\tau_{k}}}]+H_{0}\frac{\alpha D}{2e}\frac{J_{k}}{\tau_{k}}e^{-\frac{\Delta t}{\tau_{k}}}(1-e^{\frac{\Delta t}{\tau_{k}}})+\frac{e^{-\frac{\Delta t}{\tau_{k}}}}{\tau_{k}}\varepsilon_{rki}^{n}\}$$

## 3. Simulation Results and Accuracy Evaluation

An experimental facility comprising a typical reservoir-pipe-valve system was considered to verify the effects of viscoelasticity friction on the pressure head under transient flow at different water temperatures. The experimental facility is composed of three parts: a constant pressure tank, an LDPE pipe, and an end quick-closing valve, as shown in Figure 3. The total length of the pipeline is 43.1 m, the inner diameter is 41.6 mm, and the wall thickness is 4.2 mm. Both ends of the pipe section are anchored using fixed brackets. The volume of the pressure vessel is 9 m^3^. The vessel is equipped with heating and water temperature control. The transient events are caused by the closure of the downstream quick valve, and the valve closing time is 12 ms. The experimental data were reported by Gally et al. [29,33] at four different water temperatures, as summarized in Table 1. The temperature refers to the water temperature in the pipes; the initial velocity is the average velocity of the section under steady flow conditions; the reservoir pressure is the piezo-metric head of a constant pressure tank in the steady-state condition; the Reynolds number is a dimensionless value that determines the flow state of the fluid in the pipes; and the Darcy–Weisbach friction factor is a coefficient that is used to calculate the on-way resistance along the pipeline. In the numerical simulations of transient flows, the pipeline is divided into 64 elements of equal length and the piezometric head is calculated by MATLAB\2014b. The pressure wave speeds obtained by calibration in Cases 1, 2, and 3 are 370–390 m/s, and the corresponding values in Case 4 are 450–470 m/s.

The creep compliance, *J*(*t*), is calculated using dynamic tests performed on a Rheovibron apparatus [29,33]. The results for this creep parameter in LDPE at different water temperatures are listed in Table 2. *J_k_* and *τ_k_* are the coefficients of the constitutive equation of the viscoelastic pipe (i.e., Equation (9)).

### 3.1. Simulation Results for the Pressure Head

Four experimental cases [33] were analyzed, and the main difference between these cases was the water temperature in the transient flow. The water temperature affects not only the density, but also the kinematic viscosity. To analyze the accuracy of the different friction models, numerical results were calculated using the QF, no-friction (NF), and UF models.

Figure 4 presents a comparison of the experimental and simulation results at four water temperatures. The QF, NF, and UF models have strong similarities in terms of the period and phase of pressure fluctuation. However, with respect to the peak pressure values, there are some distinctions between the different friction models at the four water temperatures. At 13.8 °C, the peak pressures calculated by the models are higher than the experimental data. At 25, 31, and 38.5 °C, all of the friction models can predict the peak pressure damping of the transient flow. However, for the maximum value in terms of the peak pressure, i.e., 38.5 °C (Figure 4), the difference between the simulation and experimental results is greater than that in the other three cases. The periods of the pressure curves have clear distinctions under different conditions, and these differences are greatest at 38.5 °C.

For the peak pressures at different periods, the degree of matching between the simulation and experimental results is not perfect. As we can see from Figure 4a, the simulated trough of pressure fluctuation is lower than the experimental pressure because of the inaccuracy of the coefficients of the creep function. We will discuss this in the following section. However, the pressure attenuation of the different friction models is similar above 13.8 °C.

### 3.2. Simulation Results for the Strain

To further investigate the performance of different friction models at different water temperatures, the retarded strain based on Equations (13) and (23) was simulated. Owing to the large differences in pressure between the simulated and experimental results at 13.8 °C and 25 °C, the variations in the total strain and retarded strain simulated by different friction models are shown in Figure 5. It can be seen that at 13.8 °C and 25 °C, the attenuation of the total strain is greater than that of the retarded strain. That is because the total strain is the sum of the instantaneous strain and retarded strain. When the peak pressure rapidly decays, it results in a dramatic decrease in instantaneous strain. The total and retarded strains obtained by the QF model are higher than those obtained by the UF model. This change is most pronounced at the beginning of the transient flow. However, in the latter stages of the transient flow, the strains of the QF and NF models gradually approach each other. This is also consistent with the conclusions reported by Duan [34]. In addition, it can be observed that the maximum strains of the QF model are larger than those of the UF model. In particular, for each case in Figure 5, the total strain and retarded strain exhibit a greater delay at higher water temperatures.

### 3.3. Accuracy of the Numerical Results

#### 3.3.1. Mean Absolute Percentage Error

In Figure 4, the QF and NF models are very similar above 25 °C, particularly in the description of the pressure peaks and phases. The MAPE reflects the accuracy of the peak pressure damping simulations. To assess the effects of the friction of transient flows at different water temperatures, the MAPE evaluation method was carried out as follows:(24)$$\mathrm{MAPE}=\frac{\sum_{i=1}^{k}\frac{\left|H_{i,num}-H_{i,\exp}\right|}{H_{i,\exp}}}{k}\times 100\%$$
where $H_{i,num}$ is the *i*th simulated peak and valley pressure, $H_{i,\exp}$ is the *i*th experimental peak and valley pressure, and *k* is the number of pressure extremums.

Figure 6 shows the differences in the peak pressure damping calculated by the QF and UF models for the four cases. The percentage of deviation between the peak pressure calculated by the UF model and the corresponding experimental value at 13.8 °C is approximately 22.94%; the percentage of corresponding difference in the QF model is 31.97%. Similarly, the difference in MAPE values calculated using the 1D friction models at 25 and 31 °C is approximately 10% (i.e., at 25 °C, QF model: 18.54%, UF model: 7.59%; at 31 °C, QF model: 17.12%, UF model: 9.42%). However, at 38.5 °C, there was a slight deviation (i.e., QF model: 3.14% and UF model: 3.6%).

Combined with the pressure curve in Figure 4, the MAPE values were analyzed. At 13.8 °C, the peak and valley values of pressure fluctuation obtained by the QF and UF models display large differences relative to the experimental ones. Thus, these differences result in a large MAPE value. At 25 °C, there is no significant difference between the pressure peak values of the two models and the experimental pressure peak values. However, with respect to the pressure valley value, the simulation results of the UF model are closer to the experimental results than those of the QF model. This is the reason that the MAPE value of the QF model is greater than that of the UF model, as shown in Figure 6b. The variability of pressure peak and valley values at 31 °C is similar to that at 25 °C. Thus, from the simulation accuracy in terms of pressure peak attenuation (Figure 4b,c), the MAPE value is acceptable, with it being in the range of 20%. This illustrates that the unsteady-state component contributes to a lesser degree to the difference in the numerical calculation of the pressure at 38.5 °C.

Figure 6b also shows a comparison of the peak pressure damping results simulated by the QF and UF models at 25 °C. The calculation results of the QF and UF models are coincident; however, they are higher than the experimental results. This means that the unsteady-state component has little effect on the peak pressure damping.

On the other hand, the creep compliance coefficients and retarded times of “creep function” have also been shown to affect the peak and valley pressure of transient flow in viscoelastic pipes [35]. We studied the modification of the creep curve and pressure curve when the coefficient varies at 25 °C, referring to the method proposed by Urbanowicz [35]. The detailed values of the quantitative analysis are summarized in Table 3.

The Two Tests were analyzed in detail as follows:

Test A: There were variations in term of *J_k_* values only and constancy in terms of the *τ_k_* values as well as initial *J*_0_ values. The modified function coefficients *J_k_* in case 01 were obtained by multiplying the corresponding coefficients in case 00 by 1.3, and the function coefficients *J_k_* in case 02 were multiplied by the corresponding coefficients in case 00 by 0.7.

Test B: There was an increase and decrease in terms of *τ_k_* values and constancy in terms of the *J_k_* values. To obtain function coefficients in case 03, the retarded time, *τ_k_*, of function coefficients in case 00 was multiplied by the value 0.1. On the contrary, the function coefficients in case 04 were obtained by multiplying the initial *τ_k_* of function coefficients in case 00 by 10.

Figure 7 shows the simulation results for Tests A and B, which were obtained by modifying the function coefficients in cases 01, 02, 03, and 04 at 25 °C. The simulation pressure in case 02 has lower valley values and higher peak values compared to that in case 01. In contrast with cases 01 and 02, the pressure decays faster in cases 03 and 04. From the obtained results (Figure 7), it can be seen that the variations in creep compliance coefficients and retarded time result in a mismatch between the numerical simulation and experimental results.

#### 3.3.2. Dimensionless Difference Value

To further analyze the variation in the peak pressure damping under different water temperatures, the dimensionless forms of the pressure and time was used. The dimensionless pressure head was defined as $H^{*}=(H-H_{0})g/aV_{0}$, and the dimensionless time was expressed as $t^{*}=at/L$. 

The difference evaluation method facilitates further analysis of the peak pressure and averages over a half-cycle period. The dimensionless pressure curve is divided by the centerline of the dimensionless pressure $\left(H^{*}=0\right)$ into a few half-cycles, and the interval of dimensionless time within each period is |*t**| $\left(t_{i}^{*}\leq t^{*}<t_{i+1}^{*},i=1,\cdots ,m\right)$. The percentage dimensionless difference value, *E*, is calculated as follows [36]:(25)$$E=\frac{1}{t_{m+1}^{*}-t_{1}^{*}}\sum_{i=1}^{m}W_{i}A_{i}\times 100\%$$
(26)$$W_{i}=\max{\left(|H_{i,num}^{*}-H_{i,\exp}^{*}|\right)}_{t_{i}^{*}}^{t_{i+1}^{*}};A_{i}=\int_{t_{i}^{*}}^{t_{i+1}^{*}}|H_{i,num}^{*}-H_{i,\exp}^{*}|dt^{*}$$
where $H_{i,num}^{*}$ is the *i*th simulated dimensionless peak and valley pressure, $H_{i,\exp}^{*}$ is the *i*th experimental dimensionless peak and valley pressure, *m* is the number of halfcycles, and $\max{\left(\;\right)}_{t_{i}^{*}}^{t_{i+1}^{*}}$ is the maximum value in the period $\left(t_{i}^{*}\leq t^{*}<t_{i+1}^{*},i=1,\cdots ,m\right)$.

Figure 8 presents a comparison of the dimensionless difference value, *E*, at different water temperatures. The different values in terms of *E* calculated by the QF and UF models are similar (approximately 0.027%) at 38.5 °C. At 13.8 °C, the maximum *E* values of the QF and UF models are due to the difference between the simulated pressure peak and valley values and the experimental results. At 31 °C, the larger *E* values of the QF model are caused by the difference between the simulated pressure valley value and the experimental results. The deviation may be caused by improper experimental operation, the external ambient temperature, or a specific historical accumulation of stress in the material itself. Therefore, the *E* values are acceptable, with them being in the range of 0.4%.

## 4. Analysis and Discussion

### 4.1. Energy Analysis of the Numerical Results

To quantitatively describe the contribution of viscoelasticity and friction to the peak pressure damping from the perspective of energy dissipation and transfer, energy analysis was used. According to the literature [36], the energy relation for the governing equations of transient flows can be expressed as follows:(27)$$\frac{dM}{dt}+\frac{dG}{dt}+D_{f}+W_{R}+W_{L}=0$$
where M(t) is the total internal energy, G(t) is the total kinetic energy, $D_{f}$ is the total rate of frictional dissipation, $W_{R}$ is the power on the pipe wall, and $W_{L}$ is the power from the ends of the pipeline.

The total rate of frictional dissipation, $D_{f}$, power on the pipe wall, $W_{R}$, and the rest term of Equation (27) are calculated as follows:(28)$$D_{f}(t)=\frac{4}{D}\int_{0}^{L}\tau_{w}(x,t)Q(x,t)dx$$
(29)$$W_{R}(t)=\int_{0}^{L}\rho gH(x,t)q_{R}(x,t)dx$$
(30)$$M(t)=\frac{\rho g^{2}A}{2a^{2}}\int_{0}^{L}H^{2}(x,t)dx$$
(31)$$G(t)=\frac{\rho }{2A}\int_{0}^{L}Q^{2}(x,t)dx$$
(32)$$W_{L}(t)=\rho g[H(L,t)Q(L,t)-H(0,t)Q(0,t)]$$
where $q_{R}=2A\partial \varepsilon /\partial t$ is the radial flow per unit pipe length.

#### 4.1.1. Analysis of Viscoelasticity and Friction

Figure 9 shows the numerical results for the work performed by the viscoelastic term and the friction dissipation rates of the UF model in the four cases. As shown in Figure 9, the rate of work on the pipe wall due to the viscoelastic effect shows a cyclical fluctuating trend, and it has both positive and negative values. When the transient pressure head increases, the fluid performs positive work on the pipe wall. At this time, part of the energy in the viscoelastic pipe is dissipated as heat energy, whereas the other part is stored in the pipe owing to the viscoelastic characteristics of the pipe. When the pressure head decreases, the pipe wall performs negative work on the fluid. These results are consistent with those of previous studies [37,38]. In particular, for each case shown in Figure 9, the rate of work on the pipe wall owing to the viscoelastic effect has a greater delay at higher water temperatures.

The work performed by frictional dissipation is always positive, and its value is reflected in the corresponding peak pressure damping. The work performed by the friction is larger than that performed by the viscoelasticity in the first stage of pressure fluctuation (dimensionless time = 0 to *L*/*a*), which implies that the maximum peak pressure of the transient flow (Figure 4) is mainly affected by friction. In the subsequent pressure fluctuation, the work performed by the viscoelasticity is larger than that performed by the friction.

#### 4.1.2. Analysis of the QF and UF Models

Figure 10 shows the dissipation rates of the transient flow in the QF and UF models at 13.8, 25, 31, and 38.5 °C. The results show that in the first stage of the transient flow, there is no significant difference between the friction effects of the QF and UF models in the four cases. In the subsequent stages of the transient flow, there is a trend of rapid decay in the work performed in the results of the UF and QF models. The decay trend of the work performed in the results of the QF model is steeper than that of the UF model. This indicates that the UF component has a negligible influence on the peak pressure damping of the transient flow in viscoelastic pipes.

### 4.2. Discussion of the Contribution of the Viscoelastic Component at Different Water Temperatures

In the following quantitative analysis, to distinguish the effect of viscoelasticity from the QF and UF components, the contributions of the QF, UF, and viscoelastic (VE) components to the pressure damping are defined as follows [19]:(33)$$\gamma_{p}=\frac{\Delta H_{p}^{T}-\Delta H_{p}^{S}}{\Delta H_{p}^{T}}\times 100\%$$
where *γ* is the contribution of the pressure damping, Δ*H* is the total pressure head attenuation relative to the initial transient pressure head, the superscript *T* indicates experimental results, superscript *S* indicates the simulated pressure, and subscript *p* indicates the point at the pressure peak.

Figure 11 shows the contributions of the QF + VE, UF + VE, and VE components of the transient flow model to the peak pressure damping in the four cases. As shown in Figure 10, the contribution to the peak pressure damping is the greatest in the case of the transient flow model with the UF and VE components. The minimal contribution to the peak pressure damping under transient flow occurs when only the VE component is considered. At 13.8 °C, the contribution of the UF component is greater than that of the QF component (with a difference of approximately 0.3%). However, in the other three cases, the contribution of the QF component gradually approaches the corresponding value of the UF component (with a difference of approximately 0.05%). This indicates that a higher water temperature weakens the effect of the UF component on the peak pressure damping. This is because the variation in water temperature affects the density and viscosity of the liquid, which influences the calculation of the unsteady friction near the pipe wall. On the other hand, as the water temperature increases, the creep function coefficients enlarge, which leads to extreme pressure damping [29,35].

## 5. Conclusions

This study employed the K–V model with QF and UF models to reveal the influence of viscoelastic and frictional factors on the peak pressure damping of transient flows at different water temperatures. First, the accuracy of the QF and UF models was discussed using MAPE and the dimensionless difference value, *E*. Then, energy analysis was used to analyze the rate of work due to viscoelastic and frictional factors at different water temperatures. Finally, the contribution of the pressure damping, γ, was used to discuss the effect of viscoelasticity and friction on the peak pressure damping. The main findings and conclusions are as follows.

(1)The MAPE and dimensionless difference values of the UF model at 13.8 °C are smaller than those of the QF model. However, there is a slight discrepancy in the simulation results between the QF and UF models at 25, 31, and 38.5 °C.(2)Higher temperatures result in a greater delay in the work performed by the viscoelasticity and a decrease in the frequency of work performed by the viscoelasticity. The positive and negative variation trends in the work performed by the viscoelasticity are consistent with the increasing and decreasing trends in the pressure. At a given water temperature (25, 31, or 38.5 °C), the effect of the viscoelasticity on the peak pressure damping is dominant relative to the friction.(3)The maximum value and variability of the work performed by the QF and UF models at different temperatures exhibit slight differences. The work performed by the QF and UF models decreases dramatically as the dimensionless time increases.(4)At higher temperatures, the VE component plays a more important role than the frictional component in the transient model for peak pressure damping. There is no significant difference in the contributions obtained by the QF and UF components in the model (the difference in the contribution rates is approximately 0.05%, and at 13.8 °C the difference in the contribution rates is approximately 0.3%).

## Figures and Tables

**Figure 1 materials-15-04904-f001:**
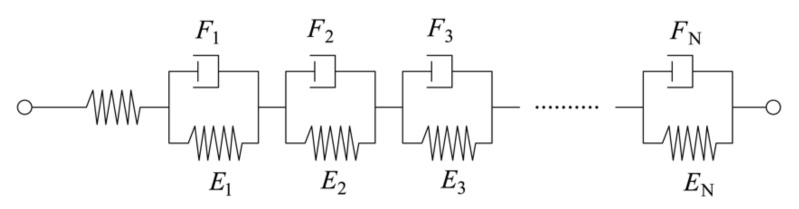
Kelvin–Voigt model.

**Figure 2 materials-15-04904-f002:**
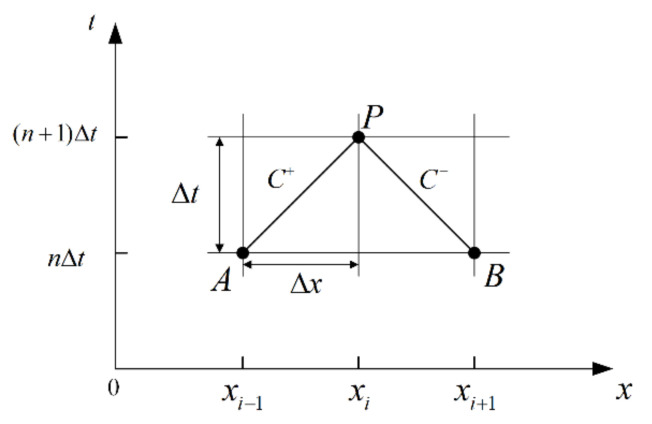
Rectangular grid system.

**Figure 3 materials-15-04904-f003:**
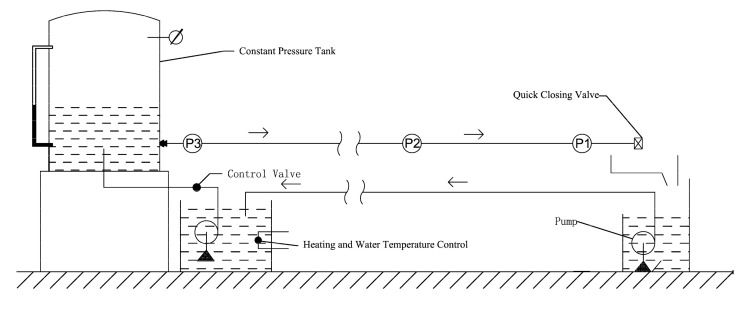
Schematic diagram of the experimental setup [29].

**Figure 4 materials-15-04904-f004:**
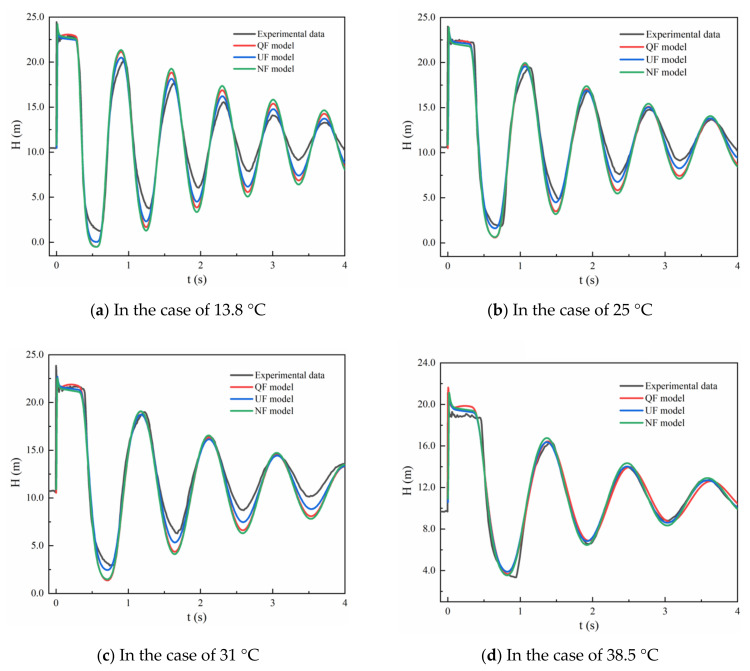
Pressure heads calculated by different friction models at four water temperatures: (**a**) 13.8 °C; (**b**) 25 °C; (**c**) 31 °C; (**d**) 38.5 °C.

**Figure 5 materials-15-04904-f005:**
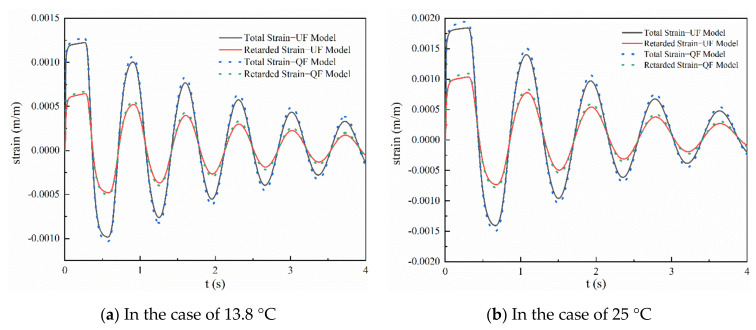
Comparison of total and retarded strains at different water temperatures: (**a**) 13.8 °C; (**b**) 25 °C.

**Figure 6 materials-15-04904-f006:**
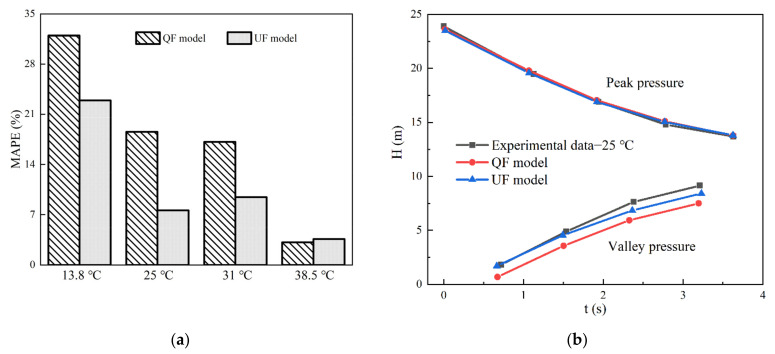
Comparison of the peak pressure damping results computed using the QF and UF models: (**a**) MAPE values; (**b**) peak pressure damping at 25 °C.

**Figure 7 materials-15-04904-f007:**
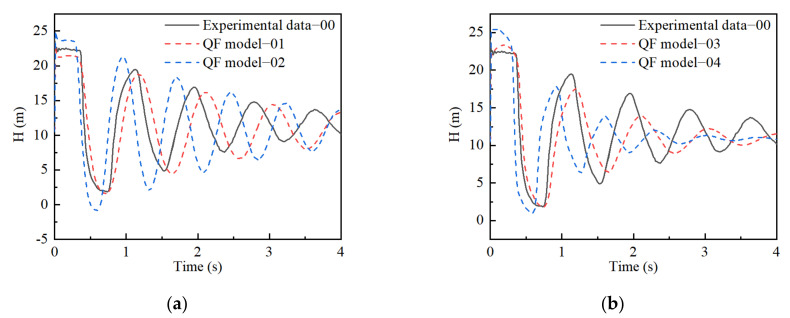
Pressure head and creep function: (**a**) Test A of QF model; (**b**) Test B of QF model; (**c**) Test A of UF model; (**d**) Test B of UF model; (**e**) Creep function of Test A; (**f**) Creep function of Test B.

**Figure 8 materials-15-04904-f008:**
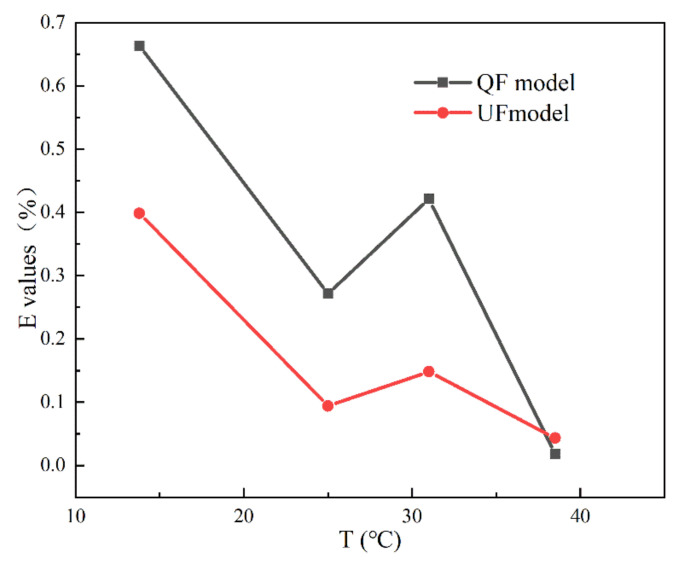
Comparison of dimensionless difference values at different water temperatures.

**Figure 9 materials-15-04904-f009:**
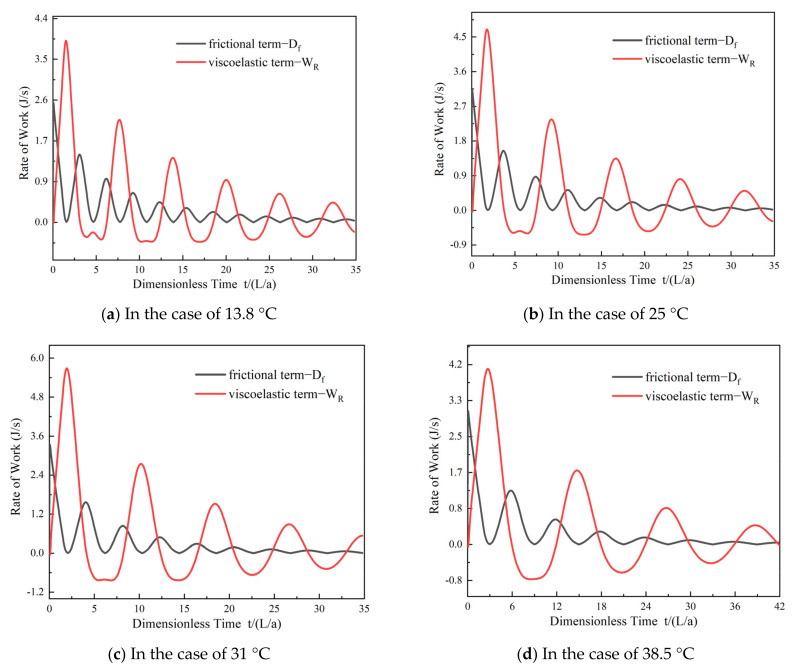
Energy results for the total rate of frictional dissipation and power on the pipe wall at different temperatures: (**a**) 13.8 °C; (**b**) 25 °C; (**c**) 31 °C; (**d**) 38.5 °C.

**Figure 10 materials-15-04904-f010:**
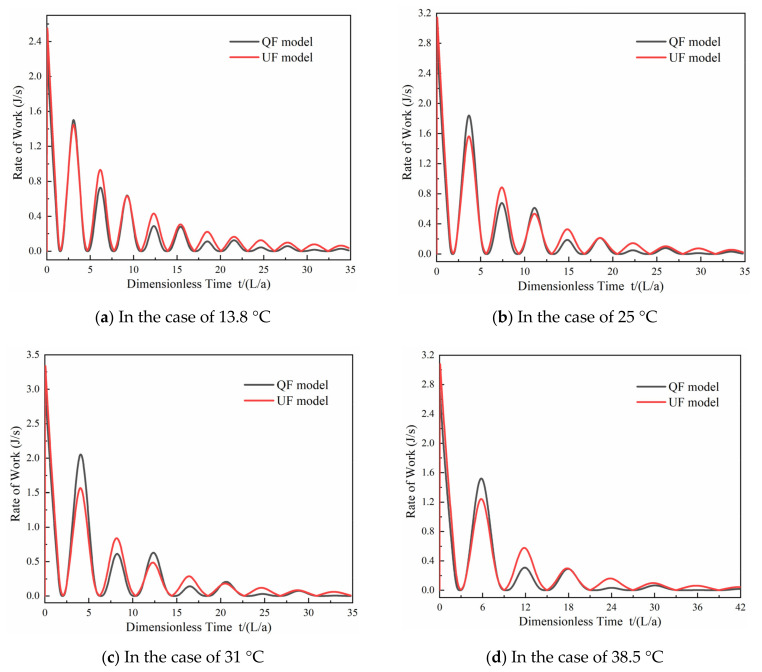
Comparison of the dissipation rates of the QF and UF models: (**a**) 13.8 °C; (**b**) 25 °C; (**c**) 31 °C; (**d**) 38.5 °C.

**Figure 11 materials-15-04904-f011:**
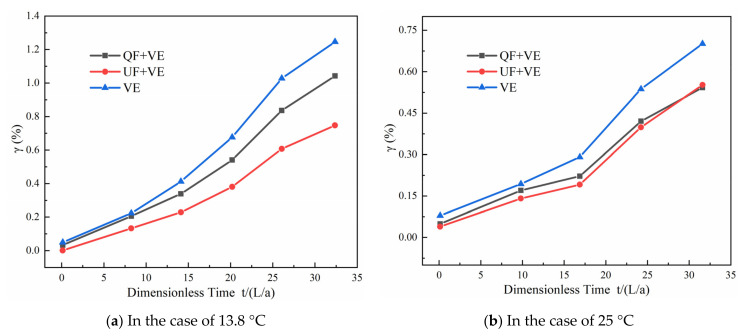
Comparison of the contributions of QF, UF, and VE components: (**a**) 13.8 °C; (**b**) 25 °C; (**c**) 31 °C; (**d**) 38.5 °C.

**Table 1 materials-15-04904-t001:** Experimental data for different cases [29].

Test	Temperature (°C)	Reservoir Pressure (×10^5^ Pa)	Initial Velocity (m/s)	Pressure Wave Speeds (m/s)	Water Kinematic Viscosity (×10^−6^ m^2^/s)	Reynolds Number	Water Bulk Modulus (×10^9^ Pa)	Water Density (kg/m^3^)	Darcy–Weisbach Friction Factor
Case 1	13.8	1.0593	0.49	379.4	1.17	17,422	2.14	999.3	0.0268
Case 2	25	1.0661	0.55	369.4	0.892	25,650	2.24	997.1	0.0244
Case 3	31	1.0670	0.57	381.2	0.784	30,245	2.27	995.3	0.0234
Case 4	38.5	1.0649	0.56	470.9	0.675	34,513	2.295	992.6	0.0227

**Table 2 materials-15-04904-t002:** Creep function coefficients in different cases [29].

Test	T(℃)	*J*_0_ 10^−9^ Pa^−1^	*J*_1_ 10^−9^ Pa^−1^	*J*_2_ 10^−9^ Pa^−1^	*J*_3_ 10^−9^ Pa^−1^	*τ_k_*_1_ 10^−4^ s	*τ_k_*_2_(s)	*τ_k_*_3_(s)
Case 1	13.8	1.414	0.516	0.637	0.871	0.56	0.0166	1.747
Case 2	25	1.542	0.754	1.046	1.237	0.89	0.0222	1.864
Case 3	31	1.791	1.009	1.397	1.628	1.15	0.0221	1.822
Case 4	38.5	2.239	1.479	2.097	3.57	1.24	0.0347	3.077

**Table 3 materials-15-04904-t003:** Tested creep function details.

Case	*J*_0_ 10^−9^ Pa^−1^	*J*_1_ 10^−9^ Pa^−1^	*J*_2_ 10^−9^ Pa^−1^	*J*_3_ 10^−9^ Pa^−1^	*τ_k_*_1_ 10^−4^ s	*τ_k_*_2_(s)	*τ_k_*_3_(s)
00	1.542	0.754	1.046	1.237	0.89	0.0222	1.864
01	1.542	**0.9802**	**1.3598**	**1.6081**	0.89	0.0222	1.864
02	1.542	**0.5278**	**0.7322**	**0.8659**	0.89	0.0222	1.864
03	1.542	0.754	1.046	1.237	**0.089**	**0.00222**	**0.1864**
04	1.542	0.754	1.046	1.237	**8.9**	**0.222**	**18.64**

## Data Availability

All relevant data are included in the paper. The data presented in this study are available on request from the corresponding author.

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
