# Peer review of "Numerical Analysis of Transient Pressure Damping in Viscoelastic Pipes at Different Water Temperatures"

_materials, 2022, doi:10.3390/ma15144904_

Round 1
Reviewer 1 Report
- Each original equation that we used to derive the mathematical modelling need to have a reference. There are several original equations in the manuscript such as equations (3), and (4), and others don't have references. Because the equations are not originally derived by the authors.
- Please state the software that you used for the numerical simulation and results.
- In the results and discussion sections, the authors have shown many results in Tables 1 and 2. However, the authors only stated what is obviously seen in the table. The authors have not discussed the physical meaning of those values shown in the table. The manuscript will be read not only by people in the same field but people in the different fields also will read the manuscript. Hence, every value or graph needs to be explained in detail so that the knowledge can be transferable to others.
- Page 6 line190: The authors stated the degree matching for simulation and experimental results is not perfect. Please elaborate why that happen.
- Page 8 line 219-221: Is the percentage of MAPE shown here is acceptable? What is the range of acceptable percentage of MAPE? and how the author calculated the difference between 1D friction models for different temperature is approximately 0.6%?
- For Figure 6(b), the numerical results for QF and UF are higher and the gap (difference) with the experimental results is seems quite big. Is it acceptable? Please clarify this.
- Page 9 line 247-248: From Figure 7, it seems that QF model is not accurate for lower temperature, but in this sentence mention UF model's inaccuracy instead of the inaccuracy of QF model. Why the author does not discussed the QF model's inaccuracy?
- There are a lot crucial findings in the manuscript, please include in the abstract.
- To make the manuscript readable, please double-check the grammatical error in the manuscript.
- If any figure from the internet or other manuscript is taken, please cite the sources.
Author Response
We are very grateful for the reviewer's suggestions, and we have attached the specific modifications.

Reviewer 2 Report
The reviewed paper is interesting, and can be published after introducing minor revision. My detailed comments are added as note windows in the pdf file - so please study them carefully when working on the revision and answers for the reviewers.

Author Response

(The authors gave the same response as above.)
